# Extraction Method Affects Contents of Flavonoids and Carotenoids in Huanglongbing-Affected “Valencia” Orange Juice

**DOI:** 10.3390/foods10040783

**Published:** 2021-04-06

**Authors:** Qili Li, Tao Li, Elizabeth A. Baldwin, John A. Manthey, Anne Plotto, Qun Zhang, Wei Gao, Jinhe Bai, Yang Shan

**Affiliations:** 1Hunan Agricultural Product Processing Institute, Hunan Academy of Agricultural Sciences, Changsha 410125, China; liqili626@163.com (Q.L.); litao209404@163.com (T.L.); zqun208@163.com (Q.Z.); 2Hunan Province Key Lab of Fruits & Vegetables Storage, Processing, Quality and Safety, Changsha 410125, China; 3Hunan Province International Joint Lab on Fruits & Vegetables Processing, Quality and Safety, Changsha 410125, China; 4USDA-ARS, Horticultural Research Laboratory, Fort Pierce, FL 34945, USA; baldwinliz052@gmail.com (E.A.B.); john.manthey@usda.gov (J.A.M.); anne.plotto@usda.gov (A.P.); 5Inspection and Testing Center of Quality and Measurement, Yueyang 414000, China; gaowei609@163.com

**Keywords:** orange juice, shear force extractor, reamer extractor, flavonoids, carotenoids

## Abstract

A previous study using healthy “Valencia” orange fruit found that juicing extraction methods dramatically changed the orange juice (OJ) flavor and phytochemical profiles. The present study was conducted to confirm whether the same changes occur when Huanglongbing (HLB)-affected oranges were used. HLB has extensively spread to most OJ processing regions around the world, substantially deteriorating fruit and juice flavor quality and altering the phytochemical profiles. In this study, the effect of two major juice extractor types, a shear force extractor (SFE) and a reamer extractor (RE), on the juice quality and physiochemical profile was assessed using HLB-affected orange fruit. Juice extracted via SFE resulted in a lower yield with lower peel oil and higher pellet (peel tissue particles) content compared to juice obtained via RE. The SFE juice also had higher levels of hesperidin and other flavonoid glycosides, mainly due to plentiful peel tissue particles. The SFE juice was also abundant in carotenoids due to a large amount of flavedo particles in the juice. On the other hand, polymethoxylated flavones occurred at higher concentrations in the RE juice, and this may be due to the higher peel oil content in this juice. The SFE juice was rich in flavonoid glycosides and carotenoids, which are associated with potential antioxidant properties; however, the extra portion of the phytonutrients mostly existed in the pellets and possibly had low bioavailability. The results obtained from the HLB-affected oranges are in agreement with the previous observations of healthy oranges.

## 1. Introduction

Huanglongbing (HLB) is a citrus disease caused by *Candidatus* Liberibacter spp. and is threatening citrus industries in Asia, Africa, and America [1,2]. Previous studies have demonstrated that HLB-affected orange fruit and juice are sour and bitter, with less sweetness, associated with high titratable acidity and bitter limonoids and/or low sugar content [3,4,5,6,7]. However, some research studies on HLB-affected orange juice (OJ) have also reported this juice to be rich in flavonoids—the most abundant phytochemical, induced by the stress response in plants [4,5,6,7,8]—and poor in carotenoids, while lacking an orange color [9,10]. Earlier reports of healthy oranges (prior to the widespread distribution of HLB disease) showed that juice extracted by a reamer extractor (RE), one of the two major commercial extractor types, contained much fewer flavonoid glycosides and carotenoids [11,12] compared to another major extractor type—a shear force extractor (SFE), which crushes some of the peel tissues into small particles, while separating most of the oil from the juice as it is extracted. Thus, potentially abundant flavonoid glycosides in albedo, and carotenoids in flavedo tissues, are introduced into the juice by the incorporation of peel particles [11,12]. On the other hand, after juice extraction by RE, the peel remains mostly intact and only the inner flesh is removed [11,12]. Thus, few peel particles are introduced into juice. However, when the RE method was used in the preparation of fresh OJ, peel oil from the oil glands in the peel tissue was squeezed into the juice, dramatically increasing the peel oil content, as well as the content of polymethoxylated flavones which are dissolved in peel oil [11,12]. The OJ processing industry generally removes the peel oil before RE extraction to avoid negative impacts to the juice flavor [13] and to recover the oil as a valuable co-product.

Orange juice has long been known to play an important role in healthy diets, contributing to individuals’ daily intake of folic acid, vitamin C, potassium, calcium, and other minerals [14,15]. Recently, particular benefits of orange juice have been shown to be related to phytonutrients, the biologically active components, such as antioxidants [16,17,18] and fiber [19], which are helpful in reducing the risk of certain cancers, heart disease and diabetes [20,21,22,23], the leading causes of disability and death. Recent research has shown that enhanced flavonoids, especially the abundant hesperidin and naringenin; carotenoids, such as β-cryptoxanthin, lutein, and fiber content, directly affect the health benefits associated with OJ [24,25].

Flavonoids in OJ generally contribute astringency and bitterness, and their negative influence became obvious in HLB OJ [6,7,8]. The off-taste in HLB “Valencia” OJ was associated with increases in hesperidin, nobiletin, tangeretin, narirutin and didymin [26]. Sugars and organic acids were frequently associated with HLB severity [4,5], but not closely correlated with the overall satisfaction and sensory perception of orange juice—rather, flavonoids, terpenoids, and volatile aromas play important roles in improving consumer overall satisfaction [27]. Dala-Paula et al. [28] suggested that hydroxycinnamates also contribute to the bitterness of HLB OJ. Limonoids, limonin and nomilin, are the major contributors to bitterness in HLB OJ [4,5,6,7,8], and previous experiments have shown that they behaved similarly to flavonoid glycosides in that they were much higher in SFE extracted OJ compared to RE [11]. Since increased bitterness is one of the major causes of taste quality deterioration in HLB OJ, RE should be preferred as it can prevent excessive peel particles from entering the juice stream [11,12].

Some reports have claimed that the phytonutrients in orange juice are highly bioavailable [17,29]. However, many of these compounds may precipitate during cold storage, and partially loose their bioavailability [30]. Most of the flavonoid glycosides in peel particles precipitate to pellets and become less bioavailable [30,31].

The objectives of this study were to compare the overall flavor and health-related quality attributes—especially bioactive phytochemicals—in OJ extracted by SFE and RE from HLB-affected oranges in order to confirm the results found in healthy OJ. Additionally, this study discusses the structural origin of the phytochemicals in fruit, and provides recommendations to the juice processors for juice extractor selection.

## 2. Materials and Methods

### 2.1. Standards and Chemicals

HPLC-grade ethanol, methanol, dimethyl sulfoxide (DMSO) and methyl tert-butyl ether (MTBE) were purchased from Tedia (Fairfield, OH, USA). Deionized water (18.20 MΩ cm) was produced with a Milli-Q system from Millipore (Bedford, MA, USA). Feruloyl putrescine, β-cryptoxantin, α-carotene, β-carotene, lutein, lycopene, 9-Z-violaxanthin, limonin, nomilin were purchased from Sigma-Aldrich (St. Louis, MO, USA). Feruloyl galactaric acid, vicenin-2, narirutin-4′-glucoside, hesperetin glucoside, isovitexin, vitexin, narirutin, diosmin, hesperidin, isosakuranetin rutinoside, sinensetin, nobiletin, tetramethylscuttelarien, heptamethoxyflavone, tangeretin, and 5-desmethyl nobiletin were obtained from the USDA (United States Department of Agriculture) citrus flavonoid collection from previous studies [32,33,34].

### 2.2. Fruit Source and Juice Extraction Process

“Valencia” oranges (*Citrus sinensis* (L.) Osbeck) were harvested from a commercial grove located in Jiangyong, Hunan, China on 10 May 2017. The trees were severely affected by HLB, with typical symptoms including blotchy mottled leaves, severe twig dieback and greenish, small and asymmetrical fruit.

A total of 300 kg of fruit was transported and stored at 5 °C for a total 3 days before washing and processing. The fruit were then randomly divided into two equal groups to represent two treatments, and each treatment was further divided into three biological replicates. The following two juice extractors were used: a JBT Fresh’n Squeezer Point-of-Sale Juicer (JBT FoodTech, Lakeland, FL, USA) with a shear force extractor (SFE), and a NS-2000E-6 Citrus Juicer (New Saier, Changzhou, Jiangsu, China) with reamer extractors (RE). The mechanism for SFE extraction consists of upper and lower cups, both with 25 fingers. These mesh together with a fruit in the center and a perforated strainer tube removes the central core and seeds while the juice exits the holes. The fingers shear the rind into 25 strips. For the RE juice extraction, the fruits were automatically fed into the juicer, cut in half and the halved fruits were pressed onto the automatic self-reversing reamer.

Juice samples were immediately used for physiochemical analysis or stored at −80 °C until analysis. To avoid photooxidation of carotenoids, the sample preparation was conducted in low light and cold temperature at 4 °C and samples sealed in brown bottles.

### 2.3. Separation of Supernatant and Pellet

Juice samples were centrifuged at 27,000× *g* for 30 min. Separated supernatants and pellets were frozen at −80 °C and then freeze-dried. Dried samples were stored in the dark in air-tight containers at −80 °C.

### 2.4. Insoluble Solids Analysis

Insoluble solids content (ISC) was determined by measuring the pellet after removing the supernatant from the centrifuged juice and washing off sugars and other soluble solids. Specifically, juice samples were centrifuged at 27,000× *g* for 30 min at 4 °C. Pellets were carefully resuspended in deionized water (the equivalent amount as in the original juice), and then centrifuged again under the same condition. The final pellets were collected and freeze-dried. The ratio of final dry pellet weight to original juice weight represents the ISC.

### 2.5. Viscosity

Fresh juice was used for viscosity measurement using a viscometer (Brooksfield, Middleboro, MA, USA) with a spindle No.02, and at 200 rpm. Viscosity was measured in centipoise (cP) as the mean of three replicate samples.

### 2.6. Juice Color Analyses

Juice, supernatant and pellet colors were measured for CIE L*, a*, and b* values using a Chromameter (Minolta CR-400; Konica Minolta, Osaka, Japan) and the instrument was set at illuminant C and 2° observer angle. For the pellet color, the probe was directly applied to the freeze-dried pellet surface. For the juice and supernatant samples, the liquid was poured into a 10 mL cuvette (optical path, 10 mm) and measurements were taken using the accessories for liquid samples through the cuvette.

### 2.7. Peel Oil Analysis

Peel oil content in the juice was analyzed by the bromate titration method [12]. Briefly, juice samples were distilled and limonene in the distillation was determined by titration with bromide–bromate solution. Peel oil concentration was calculated based on 90% of the peel oil being limonene [35].

### 2.8. Sugar, Acid and pH Analysis

The soluble solids content (SSC) was measured by the refractive index using a digital refractometer (ATAGO PR-101; Atago Co., Tokyo, Japan), and titratable acidity (TA) and pH were determined from a titration of 10 mL of juice with 0.1 mol L^−1^ NaOH to a pH 8.1 endpoint using a titrator (808 Titrando; Metrohm, Riverview, FL, USA). This was conducted with three replicate samples per treatment.

For analysis of individual sugars and acids, approximately 10 g of juice was extracted using 5 mL zinc acetate and potassium ferrocyanide, and brought to 100 mL with deionized water. The mixture was subjected to sonication for 30 min, then passed through a 0.45 µm PTFE filter to remove particles and microbes. Samples were analyzed using a high-performance liquid chromatography (HPLC) system (Shimadzu, Kyoto, Japan), including an LC-20A pump, a refractive index detector (RID), and a DUG-20A analytical degasser equipped with a YMC-Pack Polyamine II/S column (4.6 × 250 mm, YMC, Tokyo, Japan). The mobile phase consisted of water (solvent A) and acetonitrile (solvent B) with the following gradient: 30% of solvent A and 70% of solvent B in 40 min at a flow rate of 1.0 mL min^−1^. The injection volume was 20 μL. The separation was performed in the isocratic elution mode.

### 2.9. Flavonoid Analysis

For supernatant samples, 10 mL of 80% ethanol was added to 1 g of freeze-dried supernatant powdered samples and disrupted using a sonicator (Omni Sonic Ruptor 250, Omni International, Kennesaw, GA, USA) at a 70 pulse and 6.5 power setting for 10 min. The mixture was centrifuged at 5000× *g* for 10 min at 4 °C.

For pellet samples, 5 mL dimethyl sulfoxide (DMSO) was added to 0.1 g of freeze-dried pellets and disrupted using the abovementioned sonicator for 30 min. The mixture was centrifuged as above, and the supernatant collected. Three mL DMSO was added to the residue, and the same procedure was repeated. Supernatants from both extractions were combined. Mangiferin was added as an internal standard prior to analysis by HPLC for both the supernatant and pellet samples.

Analyses were performed on a Waters 2695 Alliance Separation Module (Waters, Medford, MA, USA), connected in parallel with a Waters 996 photodiode array (PDA) detector and a Waters/Micromass ZQ^TM^ single-quadrupole mass spectrometer, equipped with an electrospray ionization interface (ESI), based on Manthey’s method [36]. Briefly, the sample amount was 10 µL. Compound separations were achieved using a Waters XBridge C8 column (150 × 4.6 mm) with a solvent gradient program. The post-column split to the PDA and mass ZQ detector was 10:1 (*v*/*v*). MS analysis was operated in negative mode and the mass spectrometric data were collected from *m*/*z* 150 to 1600, scan rate 1 s^−1^, cone voltages 20 and 40 V. The conditions of the ESI source were as follows: ionization positive mode, ES+; desolvation gas (N2) flow-rate 465 L h^−1^, cone gas (N2) flow-rate 70 L h^−1^; capillary voltage, 3.0 kV; source temperature, 100 °C; desolvation temperature, 225 °C.

Quantification of flavonoids was made using either the ESI-MS mass-extracted total ion chromatograms (TIC) obtained in scanning mode or the single-ion response (SIR) mode. To normalize the ZQ mass spectrometer response during sequential runs, an internal standard, mangiferin, was measured at 423 *m*/*z*.

### 2.10. Carotenoid Extraction and Analysis

Sample extraction from freeze-dried supernatant and pellet powders was the same as above.

Carotenoids in the extraction were analyzed using HPLC based on Scott’s method [37]. Briefly, the injected sample volume was 20 µL and a C-30 YMC Carotenoid column (150 × 4.6 mm, 5 µm; YMC Co. Ltd., Komatsu, Japan) protected by a guard column. The mobile phase was provided by a three-solvent gradient program. The initial composition was 4:81:15 (*v*/*v*/*v*) water:methanol:methyl tert-butyl ether and this was altered with linear gradients to 4:6:90 (*v*/*v*/*v*) by 60 min at a flow-rate of 1 mL min^−1^, at 30 °C. A PDA detector was used and the scanning range was 200–700 nm at 5 nm increments. The quantification of each compound was calculated based on the standard curves obtained using the authentic standard chemicals.

### 2.11. Statistical Analysis

Results were evaluated using analysis of variance (ANOVA), and subsequently, Tukey’s HSD test using SPSS V11.0 (SPSS Inc., Chicago, IL, USA). All of the measurements were repeated three times. The significance levels were set at 0.05 and 0.01.

## 3. Results and Discussion

### 3.1. General Juice Features

The differences between SFE and RE extracted juice samples were obvious, even visually: the SFE juice was a bright orange color and the RE juice was a muted orange color in comparison (Figure 1). When the two juice types were separated into a pellet and supernatant, the color trends in the pellets were similar to those found for the OJ: bright orange for SFE and a more muted orange for RE juice pellets (Figure 1). However, the supernatant color was totally unexpected: almost colorless for SFE and a light orange color for RE supernatants (Figure 1). It is likely that the supernatant color was associated with the carotenoids dissolved in the peel oil that remained in the supernatants as an oil micro-emulsion. In SFE juice, the peel oil content was 0.22 g kg^−1^ (Table 1), which is in the preferred oil content range of 0.1–0.25 g kg^−1^ for most citrus juices [12]. As an industrial standard, the maximum USDA designated oil content for “grade A” orange juice is 0.35 g kg^−1^ [12]. However, in RE juice, the peel oil content was 1.96 g kg^−1^, about nine-fold greater than in SFE juice (Table 1). Thus, a stable micro oil drop emulsion was established in RE juice and the micro oil drops, containing the dissolved pigments, remained in the supernatants. On the other hand, in SFE juice with little peel oil, water insoluble colorants were separated from the supernatants, remaining in the pellets, thus, the pellet color was bright orange with high L* (lightness), a* (red) and b* (yellow) values (Figure 1). In commercial OJ processing using RE, it is customary to proceed with an oil removal procedure before juice extraction in order to meet the USDA standard of oil content [12,13]. For food service fresh juice production, peel oil content is not a quality consideration, and both SFE and RE can be used without prior oil removal. As mentioned in the introduction, there is a mechanism in the JBT SFE extractor to separate oil from the juice [12,13].

For other internal juice quality parameters, the RE type provided a higher yield and juice with a lower SSC and individual sugars (Table 1), indicating that RE not only pressed more peel oil, but also peel juice fluid—contained in the flavedo and albedo—into the OJ, which diluted the sugar content in the juice. As a consequence, RE juice also had low SSC/TA ratio and viscosity (Table 1). Viscosity is the consistency or thickness of a fluid due to internal friction. Therefore, the higher the SSC, pectin and ISC components in the SFE juice, the higher the internal friction, which increases the value for the viscosity. In contrast, low SSC and ISC values in the RE juice led to a low viscosity, and the high peel oil content further decreased the viscosity (Table 1). This phenomenon was only seen for RE extraction without previous oil and fluid removal, however, for industrial juice, the SSC is comparable to the SFE-obtained OJ since the peel oil is removed prior to juicing [11,12].

In comparison with RE, SFE extracted juice had 2.8-fold more ISC (Table 1). The SFE extractor introduced more ISC into the juice from the 25 cuts of the fruit rind tissues including flavedo tissue particles which imparted an orange color to the pellets (Figure 1). OJ has garnered criticism from consumers concerning the lack of fiber in comparison with orange fruit, thus, the high ISC in SFE juice is associated with increased fiber and other bioactive chemicals, as demonstrated below.

### 3.2. Flavonoids

Fifteen flavonoids were analyzed in the juice samples and the total content was 990 and 787 mg L^−1^ juice, in SFE and RE juice, respectively (Table 2, Figure 2 and Figure 3). The major flavonoid class was flavonoid glycosides (FGs), in which hesperidin (HSP) was the dominant compound, followed by vicenin-2 (VCN), narirutin (NR), diosmin (DSN), isosakuranetin rutinoside (ISR), narirutin-4′-glucoside (NR4G), hesperetin glucoside (HSPTN-G), vitexin (VTX), and isovitexin (IVTX) (Figure 2). The total FGs content was similar in supernatants from SFE and RE juice (Figure 2). However, the content in pellets was substantially different between the two processing methods: the FGs content in SFE juice pellets was comparable to the amount in the supernatant, but the FGs content in the RE juice pellets represented only about 10% of the supernatant content (Figure 2). The total FG level in the pellets from SFE juice was about 42% higher than in the pellets from RE juice (Table 2). Studies have shown that the major FGs, such as HSP, VCN and NR, are mainly concentrated in the albedo and segment membranes of orange fruit, and to a lesser extent in the flavedo, while being absent in juice vesicles [38]. We could deduce that the SFE extraction process introduced more albedo and segment membranes from the fruit into the juice. Flavonoid glycosides are relatively less hydrophobic and exhibit some water solubility [39] in orange juice. The abundant precipitation of FGs in SFE juice pellets was most likely from the membrane and albedo particles in the juice, but the low water solubility of FGs also contributed [11]. Previous data revealed the extent of FGs in precipitates of SFE extracted “Hamlin” OJ, with very high FGs, was considerable and within 4 days of storage at 4 °C, three quarters of the HSP precipitated from the supernatant to the pellets [11].

Since the bioactive flavonoids are in the aqueous phase [30,31], the practical health value of FGs in both juice samples should be similar (Figure 2 Supernatant). However, the concentration of the flavonoid—HSP in the RE supernatant—was 343.75 mg L^−1^, which was 30% higher than the 261.26 mg L^−1^ in the SFE supernatant (Figure 2 Supernatant). This complicates the determination of which juice is more nutritionally sound since the SFE juice had a greater total FG content, but the RE juice had equal or more bioactive FGs [40].

Another flavonoid class of phytochemicals is the polymethoxylated flavones (PMFs), including nobiletin (NOB), heptamethoxyflavone (HMS), tangeretin (TAN), sinensetin (SIN), tetramethylscutellarein (TMS), and 5-desmethylnobiletin (DMN) (Figure 3). The total PMF content in the RE juice was 101.46 mg L^−1^, in comparison with 16.22 mg L^−1^ in the SFE juice, a difference of more than six-fold (Table 2 and Figure 3). In both juices, PMFs were higher in the supernatant than in the pellet (Table 2 and Figure 3). The high PMF content was associated with peel oil and peel juice, which were largely introduced into the juice by the RE extractor (Table 1 and Table 2 and Figure 3). The peel oil content in the SFE juice was only about 11% that of the RE juice (Table 1). The report by Nogata et al. (2006) showed that PMFs were mainly located in the flavedo (oil), and to a lesser extent in the albedo, but were nearly absent in segment membranes and juice vesicles [38]. Therefore, we can deduce that the high content of PMFs was related to the high content of peel oil in the RE juice. In comparison with the total FG content, the total PMF content in RE was as low as <15%, and even less than 2% in SFE juice (Table 2, Figure 2 and Figure 3).

### 3.3. Carotenoids

Five carotenoids were determined in differently prepared juice supernatants and pellets (Figure 4 and Table 2): 9-Z-violaxanthin, lutein, β-cryptoxanthin, α-carotene and β-carotene. 9-Z-violaxanthin and lutein are beneficial for eye health and are strongly related to the chroma and hue values of the juice, respectively. β-carotene, α-carotene, and β-cryptoxanthin are provitamin A carotenoids and also have antioxidant activities [41,42,43].

Previous studies have shown that the carotenoids are responsible for OJ color and that processing conditions have a strong impact on the carotenoid profile in OJ [41,42]. Our results show that the total content of carotenoids in the SFE OJ was 12.03 mg L^−1^, 2.6 times higher than in the RE juice (Table 2 and Figure 4). The results were consistent with the color values shown in SFE and RE juice, with SFE juice having a more intense orange color than the RE juice (Figure 1). The carotenoid content also explained the color differences in juice supernatant and pellets of both types. The carotenoid content in SFE juice pellets was 2.8 times higher than in the RE pellets (Figure 4). In addition, the total content of β-carotene, β-cryptoxanthin, and 9-Z-violaxanthin in both the SFE and RE pellets was much higher than the total content of α-carotene and lutein (Figure 4). This result was in accordance with those obtained by Kato et al., who reported that with the transition of peel color from green to orange, the accumulation of carotenoids (α-carotene and lutein to β-carotene, β-cryptoxanthin, zeaxanthin, and violaxanthin) was observed in the flavedo, along with changes in gene expression [44].

The supernatant color was associated with the carotenoids, which were dissolved in peel oil and remained in supernatants as an oil micro-emulsion (Figure 4). The peel oil content in RE juice was about nine-fold higher than in SFE juice. In the SFE supernatant, there was not enough peel oil to dissolve the carotenoids, thus, carotenoids generally precipitated into the pellets. The results agreed with the color of the supernatant and pellet (Table 2, Figure 1 and Figure 4).

The importance of 9-Z-violaxanthin in the color of juice, supernatant and pellet samples was obvious (Figure 1 and Figure 4). Violaxanthin belongs to the class of organic compounds known as xanthophylls. These are carotenoids containing an oxygenated carotene backbone. Carotenes are characterized by the presence of two end-groups, mostly cyclohexene rings, but also cyclopentene rings or acyclic groups, linked by a long, branched alkyl chain. Xanthophylls arise by oxygenation of the carotene backbone. Thus, violaxanthin is considered to be an isoprenoid lipid molecule. Violaxanthin is a very hydrophobic molecule, practically insoluble in water, and relatively neutral [44,45]. Thus, it seems the peel oil was the key to keep 9-Z-violaxanthin in the aqueous phase to prevent its precipitation in the pellet.

## 4. Conclusions

This research investigated the effect of two different methods of juice extraction on the physical and chemical properties and phytochemical profiles of HLB OJ. To summarize, OJ extracted via SFE showed a higher ISC, associated with abundant carotenoids and FGs, which are the major phytochemicals. The SFE juice also showed higher soluble solids and higher color values (L*, a*, b*), as compared to the RE extracted juice. OJ extracted via RE showed higher peel oil content, associated with not only the abundant PMFs in both the supernatant and pellet samples, but also equal or higher carotenoids and FGs, which are believed to be more bioavailable. The results agree with the previous study conducted using healthy “Valencia” orange fruit processed using SFE or RE extractors [11]. As HLB OJ tends to contain higher FG and limonoids contents than healthy OJ, HLB OJ extracted by SFE has a greater potential to be bitter. On the other hand, HLB OJ extracted by SFE could be a high content source of health-benefit flavonoids with antioxidant activities.

## Figures and Tables

**Figure 1 foods-10-00783-f001:**
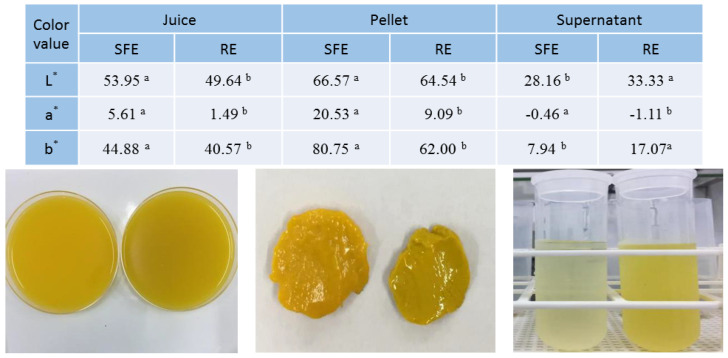
Color values (*n* = 3) and visual images of juice, pellet, and supernatant of “Valencia” orange juice processed using shear force extractor (SFE) and reamer extractor (RE) types. Color values followed by a letter indicate significant differences between SFE and RE (*p* = 0.05) within each sample type.

**Figure 2 foods-10-00783-f002:**
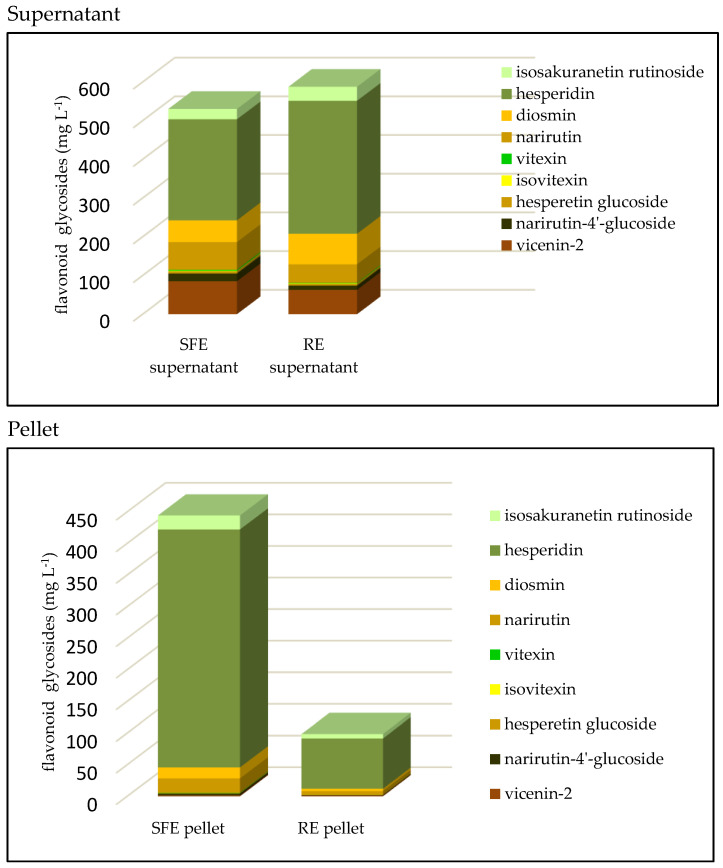
Flavonoid glycoside content in “Valencia” orange juice processed with shear force extractor (SFE) and reamer extractor (RE) types.

**Figure 3 foods-10-00783-f003:**
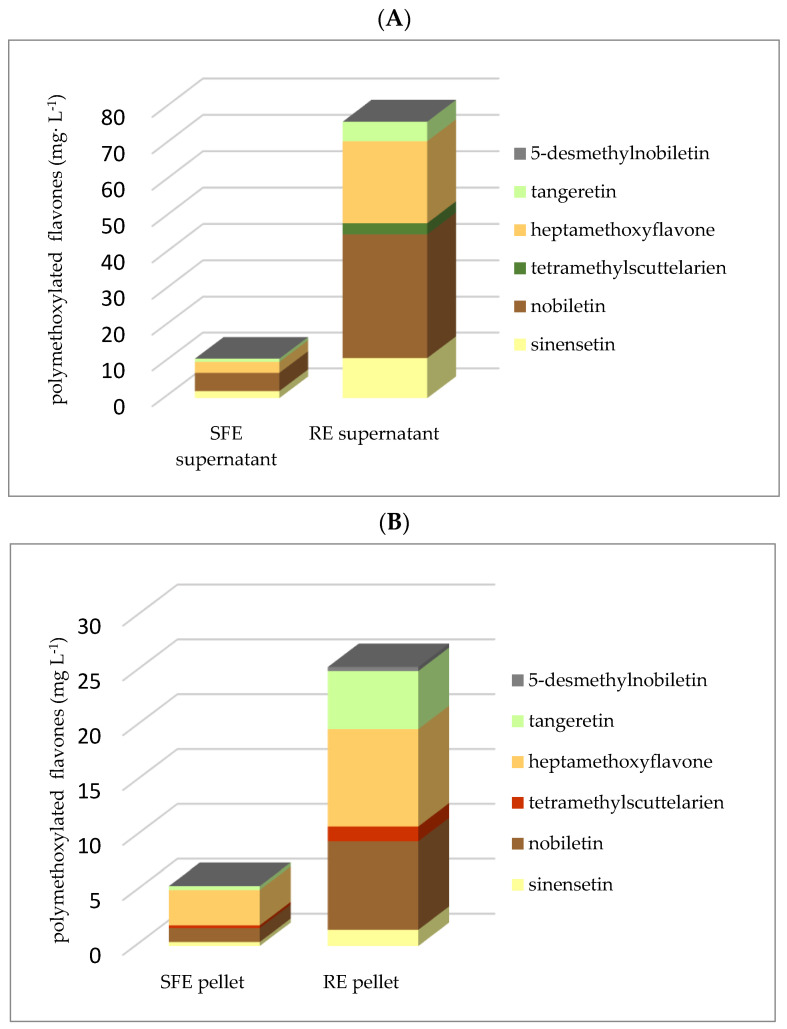
Polymethoxylated flavone content in “Valencia” orange juice processed with shear force extractor (SFE) and reamer extractor (RE) types. (**A**): supernatant; (**B**): pellet.

**Figure 4 foods-10-00783-f004:**
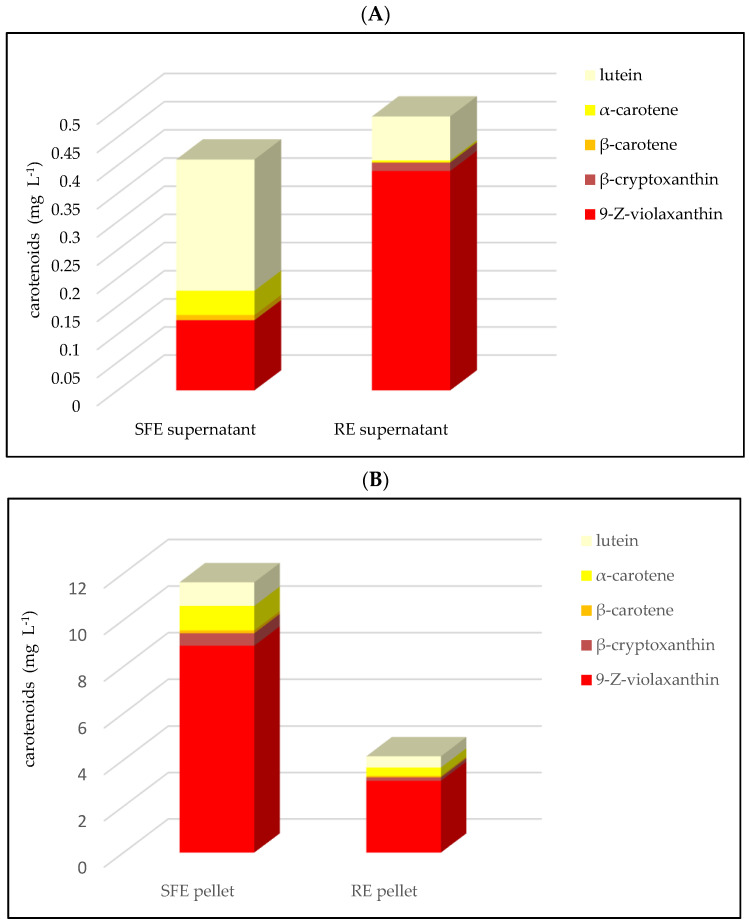
Carotenoid concentration in “Valencia” orange juice processed with shear force extractor (SFE) and reamer extractor (RE) types. (**A**): supernatant; (**B**): pellet.

**Table 1 foods-10-00783-t001:** Effect of shear force extractor (SFE) and reamer extractor (RE) type on basic juice properties: juice content, insoluble solids content (ISC), soluble solids content (SSC), titratable acids (TA), pH, SSC/TA ratio, sucrose, glucose, fructose, peel oil and viscosity in “Valencia” orange juice.

Attribute	SFE	RE	*t*-Test ^a^
juice content (%, *v*/*w*)	42.80 ± 0.67	44.60 ± 0.71	0.05
insoluble solids content (ISC, g L^−1^)	4.90 ± 0.19	1.75 ± 0.04	0.01
soluble solids content (SSC, g L^−1^)	94.3 ± 1.2	81.3 ± 2.8	0.01
titratable acidity (TA, g L^−1^)	9.1 ± 0.2	9.3 ± 0.3	NS
pH	3.89 ± 0.10	3.80 ± 0.01	NS
SSC/TA ratio	9.91 ± 0.21	7.98 ± 0.17	0.01
sucrose (g L^−1^)	27.0 ± 1.1	26.1 ± 0.9	0.05
glucose (g L^−1^)	20.8 ± 0.6	16.2 ± 1.0	0.05
fructose (g kg^−1^)	23.5 ± 0.9	19.7 ± 0.2	0.05
peel oil (g L^−1^)	0.22 ± 0.02	1.96 ± 0.12	0.01
viscosity (mPa s^−1^)	58.7 ± 1.8	46.0 ± 0.7	0.05

^a^ NS, 0.05 and 0.01 represent no significant difference or significant level at 0.05 and 0.01, respectively.

**Table 2 foods-10-00783-t002:** Effect of shear force extractor (SFE) and reamer extractor (RE) types on total flavonoid glycosides, polymethoxylated flavones, and carotenoid content in “Valencia” orange juice (mg L^−1^ juice) and the distribution between supernatant and pellet (supernatants %:pellets %).

Chemical Class	SFE	RE	*t*-Test ^a^
	(Supernatant %: Pellet %)		
flavonoid glycosides	973.32 ± 71.89 (58:42)	685.16 ± 51.64 (89:11)	0.01
polymethoxylated flavones	16.22 ± 0.84 (66.85:33.15)	101.46 ± 6.32 (75.04:24.96)	0.01
carotenoids	12.03 ± 0.54 (3.41:96.59)	4.64 ± 0.22 (10.56:89.44)	0.01

^a^ 0.01 represents significant level at 0.01.

## Data Availability

The data used to support the findings of this study are included within the article.

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
