# Peer review of "Extraction Method Affects Contents of Flavonoids and Carotenoids in Huanglongbing-Affected “Valencia” Orange Juice"

_foods, 2021, doi:10.3390/foods10040783_

Round 1
Reviewer 1 Report
The manuscript presents a study based on the comparison of two different extractors (SFE and RE) for the production of orange juices and their effect on juices' phytochemical profile and quality. The article is well written, however there some comments that have to be addressed by the authors.
- Add the extractors types in the Keywords.
- ''Discuss the structural origin (Last paragraph in the Introduction section'': No such discussion takes place in the manuscript (regarding the structural origin of the compounds). Did the authors mean the screening/identification of phytochemicals? If so, rephrase or remove this sentence.
- Section 2.8: This section describes the analysis of hydrophilic compounds (i.e., sugars and acids). PTFE filters are normally used in the analysis of hydrophobic compounds, such as carotenoids. What type of PTFE filters did the authors used?
- Section 2.10: Carotenoids extraction and analysis: No internal standard (as in the case of flavonoids) was used for carotenoids' analysis and why? The LC column implemented for the analysis of carotenoids was a C18 or a C30 column? Please provide the specifications of the column (length, diameter, particle size).
- The novelty and especially the necessity of this study needs to be emphasized in the introduction section, which it should be modified, since there are many published works related to the (phytochemical and quality) characterization of both healthy or HLB juices of Valencia variety. Did the authors elucidate any new not reported flavonoid?
https://www.frontiersin.org/articles/10.3389/fpls.2018.01976/full
https://www.sciencedirect.com/science/article/pii/S002364381830121X
https://pubs.acs.org/doi/10.1021/acs.jafc.9b07934
https://pubmed.ncbi.nlm.nih.gov/25546309/
6. As stated by the authors, one of the objectives of the study was ''compare the overall flavor and health-related quality attributes [....] to confirm the results occurred in healthy OJ''. According to other studies, it is already established that the HLB juices do not present the same quality aspects or they don't share the same phytochemical profile with the healthy juices. Thus, the focus of this study is to compare and discuss the effects of the different extractor types on HLB juices? If so, please highlight and clarify the aims of the present study in the Introduction section.
7. Please provide further info/discussion for the differences in targeted flavonoids (for example isovitexin or hesperidin glycoside) between the two different supernatants (SFE and RE) as shown in Figure 2.
8. As presented in Figure 4, why violaxanthin's content is higher in the pellets than in the supernatant of SFE compared to the other carotenoids? And why the opposite trend (higher content of violaxanthin in supernatants) is presented in the case of RE? Please support these findings with more references or possible justifications.
9. Figure 3 is missing from this version of the manuscript. Please add it and correct the word ''Scheme'' with the word ''Figure'' in the caption of the figure.
10. Please enrich the Conclusions section which is too short. In addition, as cited in the Introduction section, the authors should make some suggestions to the consumers and producers. This part is missing from the manuscript, so the authors may discuss it in the Conclusions sections.
11. Please correct the numbers in the reference list, since they are duplicated in the current form.
Author Response
Reviewer 1
The manuscript presents a study based on the comparison of two different extractors (SFE and RE) for the production of orange juices and their effect on juices' phytochemical profile and quality. The article is well written, however there some comments that have to be addressed by the authors.
- Add the extractors types in the Keywords.
We have added “shear force extractor” and “reamer extractor” in the keywords (Line 32).
- ''Discuss the structural origin (Last paragraph in the Introduction section'': No such discussion takes place in the manuscript (regarding the structural origin of the compounds). Did the authors mean the screening/identification of phytochemicals? If so, rephrase or remove this sentence.
We have added “the structural origin of the compounds” related discussion in the “Results and discussion” section, including all the major chemical groups of flavonoid glycosides (FGs, lines 274-294), polymethoxylated flavones (PMFs, lines 312-315), and carotenoids (lines 348-363).
- Section 2.8: This section describes the analysis of hydrophilic compounds (i.e., sugars and acids). PTFE filters are normally used in the analysis of hydrophobic compounds, such as carotenoids. What type of PTFE filters did the authors used?
We use PTFE filters to remove microbes and particles for all samples which will be analyzed by HPLC. The sentence has been modified to indicate the purpose (Line 153).
- Section 2.10: Carotenoids extraction and analysis: No internal standard (as in the case of flavonoids) was used for carotenoids' analysis and why? The LC column implemented for the analysis of carotenoids was a C18 or a C30 column? Please provide the specifications of the column (length, diameter, particle size).
We extracted carotenoids from freeze-dried supernatant/pellet samples (lines 188-189) and “the quantification of each compound was calculated based on the standard curves ob-tained by using the authentic standard chemicals” (lines 196-198). LC column was C30, the information, along with column length, diameter and particle size have been added (line 191-192).
- The novelty and especially the necessity of this study needs to be emphasized in the introduction section, which it should be modified, since there are many published works related to the (phytochemical and quality) characterization of both healthy or HLB juices of Valencia variety. Did the authors elucidate any new not reported flavonoid?
https://www.frontiersin.org/articles/10.3389/fpls.2018.01976/full
https://www.sciencedirect.com/science/article/pii/S002364381830121X
https://pubs.acs.org/doi/10.1021/acs.jafc.9b07934
https://pubmed.ncbi.nlm.nih.gov/25546309/
We have followed your suggestions, read the references and added the information into the Introduction and Results and Discussion sections. The major additions can be found in lines 63-74. The references added in the text are [6], [26], [27], and [28].
- As stated by the authors, one of the objectives of the study was ''compare the overall flavor and health-related quality attributes [....] to confirm the results occurred in healthy OJ''. According to other studies, it is already established that the HLB juices do not present the same quality aspects or they don't share the same phytochemical profile with the healthy juices. Thus, the focus of this study is to compare and discuss the effects of the different extractor types on HLB juices? If so, please highlight and clarify the aims of the present study in the Introduction section.
Our research objectives were to compare the overall flavor and health-related quality attributes, especially bioactive phytochemicals, in HLB OJ extracted by different juice extractors. Followed your suggestion, we enhanced the description in the Introduction section, and also enhanced the comparisons of healthy fruit/juice versus HLB fruit/juice, and effect of juice extractor on healthy OJ, thus provide reader a comprehensive understanding of cross influence among HLB and juice extractors. The improvements can be found in lines 81-85, and 372-375.
- Please provide further info/discussion for the differences in targeted flavonoids (for example isovitexin or hesperidin glycoside) between the two different supernatants (SFE and RE) as shown in Figure 2.
We have provided further info/discussion for the differences in flavonoids including the above two – line 263-294.
- As presented in Figure 4, why violaxanthin's content is higher in the pellets than in the supernatant of SFE compared to the other carotenoids? And why the opposite trend (higher content of violaxanthin in supernatants) is presented in the case of RE? Please support these findings with more references or possible justifications.
We added a paragraph (lines 354-363) to explain the hydrophobic characteristics of violaxanthin, and why it precipitated from the supernatant to the pellet, especially in SFE juice.
- Figure 3 is missing from this version of the manuscript. Please add it and correct the word ''Scheme'' with the word ''Figure'' in the caption of the figure.?
Yes. We added in the figure (lines 294-302).
- Please enrich the Conclusions section which is too short. In addition, as cited in the Introduction section, the authors should make some suggestions to the consumers and producers. This part is missing from the manuscript, so the authors may discuss it in the Conclusions sections.
Yes. We enhanced the Conclusions section – lines 371-377.
- Please correct the numbers in the reference list, since they are duplicated in the current form.
Yes. We removed the duplications, added new references and renumbered the reference list.

Reviewer 2 Report
The authors investigate the influence of extraction method on the physico-chemical characteristics of orange juice (cv Valencia) infected by Candidatus Liberibacter spp.
In general, the ms is comprehensive and well prepared. The experiments are well conducted.
In my opinion the conclusions could be more detailed.
Except this remark, I have only very minor comments to improve the quality f this work:
Abstract: 1. “Overall, the SFE juice was rich in flavonoid glycosides and carotenoids, which are associated with strong antioxidant properties, anticancer activity and rich orange color(carotenoids).” Antioxidant and anticancer activities were not evaluated therefore I suggest to remove this information.
- “mainly due to plentiful peel tissue particles,” Please insert “.” Instead of “,” at the end of this sentence.
- Materials: Indicate injection volume for LC-MS analysis of flavonoids.
- Conclusions: “The results agreed with the previous study conducted by using healthy ‘Valencia’ orange fruit.” What could be the conclusion of this observation
Round 2
Reviewer 1 Report
Please remove the word ''aqueous'' from the sentence ''higher aqueous carotenoids'' (Line 371, Conclusions). Carotenoids are lipophilic molecules with very low solubility in water.
By addressing the above stated suggestion, the revised version of the manuscript should be accepted.
Author Response
We have removed the word ''aqueous'' from the sentence ''higher aqueous carotenoids'' (Line 370, Conclusions). Please see the attachment, thank you!
